# How to Avoid Lower Priority for Smoking Cessation Support Content on Facebook: An Analysis of Engagement Bait

**DOI:** 10.3390/ijerph20020958

**Published:** 2023-01-05

**Authors:** Jezdancher Watti, Máté Millner, Kata Siklósi, Csaba Hamvai, Oguz Kelemen, Dávid Pócs

**Affiliations:** Department of Behavioral Sciences, Albert Szent-Györgyi Medical School, University of Szeged, 6722 Szeged, Hungary

**Keywords:** social media, Facebook, engagement, Facebook reaction, public health, mental health, health communication, smoker, smoking, smoking cessation

## Abstract

Facebook demotes “engagement bait” content that makes people interact. As a result of this sanctioning, public health content can reach fewer Facebook users. This study aims to determine the negative effect of engagement bait and find alternative techniques. In a three-year period, 791 smoking cessation support content was included (*n* = 791). The Facebook posts were classified into “engagement bait”, “alternative techniques” and control groups. Facebook metrics were compared between the study and control groups. The reach of Facebook page fans was significantly lower in the engagement bait group compared to the control group. On the other hand, the alternative techniques had a significantly lower rate of negative Facebook interactions, as well as significantly higher click rates compared to the control group. This is the first study to reveal the sanctioning of engagement bait on smoking cessation support Facebook posts. “Engagement bait” content has a lower ranking on the Facebook Fans’ Newsfeed page. Nevertheless, alternative techniques can circumvent the restrictions on engagement bait. At the same time, alternative techniques can stimulate the click rate and inhibit the rate of negative interactions.

## 1. Introduction

### 1.1. Content Ranking on Facebook

Facebook is one of the most popular social media platforms [1,2,3], which is widely used for public health interventions [4,5,6]. A major goal of these Facebook-based interventions could be to access many users [7]. The Facebook platform has many advantages in supporting smoking cessation. For example, it can be a useful tool to contact hard-to-reach smokers [8]; it can be more cost-effective than television advertising [9]; or it can effectively help young smokers to quit [10]. These benefits of Facebook-based smoking cessation interventions can mean even more in a country with high smoking rates, such as Hungary, where the proportion of smokers in the Hungarian adult population is 28.7% [11]. Facebook allows page administrators to increase the number of people who can see the given social media content by paying [12,13,14,15]. However, it is still unclear how public health content can reach more Facebook users in a non-paid way.

Facebook ranks all available content that can be displayed in a user’s News Feed [16,17,18,19]. If the content is ranked higher in News Feeds, it probably reaches more Facebook users, and lower-ranked content is likely to reach fewer users [18,19]. The mechanism of this algorithmic content ranking is unknown, not published, and barely researched [18,19]. However, some elements of content ranking which may determine the rank of a Facebook post in the user’s News Feed are suspected [18,20]. These elements can be the following: the performance of the given post (the rate of “shares”, “clicks” or other interactions); the Facebook user’s past activity (e.g., using the ‘page like’ button); the post type (e.g., image or video); or the timing of the published content (e.g., novelty) [18,20]. Facebook posts with a higher rate of certain interactions can reach more users [18], similar to video-based content or freshly published content. Finally, a Facebook post can be sanctioned with a lower rank in the users’ News Feed if it violates the Facebook Community Standards.

### 1.2. Engagement Bait on Facebook

“Engagement” is a widely used concept in web-based public health interventions. The engagement on Facebook can be interpreted as the usage of the public health intervention, which is divided into quantity indicators (e.g., frequency and duration of usage) and quality indicators (e.g., the use of specific buttons) [18,21]. Facebook applies the same theoretical framework of quality indicators when it provides certain interaction data (reactions, comments, shares, and clicks) to the Facebook page administrators as a group of “engagement rate” [22]. The Facebook reactions (such as “Like”, “Love”, “Haha”, “Wow”, “Sad” and “Angry”) can give an opportunity to the users to express their emotions [23]. In comments, Facebook users can publish a text or an image message related to the given content [24,25]. The “Share” button is designed to give users several ways of sending content with optional privacy settings to others [26,27]. Clicks cover any other neutral actions, such as viewing the Facebook page profile or expanding photos to full screen [28]. Lastly, there is another group of interactions, which can be interpreted as resistance against the public health intervention [18,22]. As opposed to engagement, negative Facebook interaction buttons lead to an interruption with the public health intervention [18,22]. These “anti-engagement” activities include post hides, hides of all posts, reports of spam, and the unlike of the page [18,22].

A few years ago, Facebook revealed in its Community Standards that “engagement bait” content is automatically sanctioned with a lower rank in the users’ News Feed [29]. The reason for this decision was also published: inauthentic content which leads users to interact with certain actions, such as likes, shares or comments, is not favored. [29]. Engagement bait was defined as a strategy to create Facebook posts that make people interact in order to boost engagement and achieve a greater reach on News Feeds [29]. The five subcategories of engagement bait were also defined generally as follows: react baiting, comment baiting, share baiting, tag baiting and vote baiting [29]. However, Facebook did not release more specific guidelines to avoid further manipulation of content ranking in News Feeds [29]. An exception regarding the sanctioning of engagement bait should be also mentioned [29]. The “benefit to people” content is exempt from restrictions. Facebook provides the following examples for the exception: missing child reports, raising money for a cause, or asking for travel tips [29]. It is unclear whether public health content belongs to the “benefit to people” content according to Facebook. No previous study has investigated the mechanism of engagement bait which can reduce the non-paid reach of public health content.

There is no evidence to show how the algorithmic content ranking of engagement bait works on Facebook. That is why the current research first seeks to determine the impact of engagement bait on content ranking and Facebook user activity. It should be highlighted that the recent research is an exploratory study, which analyses a set of Facebook post-level metrics and then proposes hypotheses that may then be tested in subsequent studies. The secondary research aim is to find alternative techniques instead of engagement bait that can stimulate engagement without receiving lower rankings in the user’s News Feed.

The research questions are as follows:
How do engagement bait sanctions work for promoting smoking cessation on Facebook?What is the relationship between engagement bait techniques used in Facebook-based smoking cessation intervention and users’ interactions?Are there any differences in Facebook sanctions or users’ interactions if alternative techniques are used instead of engagement bait during a smoking cessation intervention?

## 2. Materials and Methods

### 2.1. Participants

The participants were “reached” Facebook users who had seen the content of the Hungarian “CigiSzünet” (“Cigarette Break”) Facebook page. This smoking cessation intervention is run by the authors of the manuscript and other experts, students, and university lecturers at the University of Szeged. The goal of the intervention is to support smoking cessation, and not to prevent smoking. The intervention started on 2017, March 7, and the number of reached people has steadily increased. We published Hungarian language content every day or every two days. The site avoids intimidating and judgmental content, and, instead, it seeks primarily to support smoking cessation based on the Motivational Interviewing counseling approach [22,30,31]. Based on the delivery of health information, this Facebook page is a web-based smoking cessation intervention. Based on the therapeutic method, this intervention is behavioral counseling for smoking cessation with a motivational interviewing approach. Our researcher identity is transparent on the Facebook page. We keep Facebook users informed about the current research and its results.

The demographic data of the users reached (sex, age, location) is made available to the page administrators by Facebook. The summarized and anonymized page-level data were exported from the “Facebook Insights” database on 25 June 2021. These page-level data are subject to Facebook’s privacy policy, and they are provided to the page administrators by Facebook with the users’ consent. On the day of data export, 10,439 people liked the page. Of them, 53% were women and 47% were men, 1% were between 13 and 17 years old, 80% were between 18 and 35 years old, and 19% were older than 35 years. Ninety-five percent of them indicated Hungary as their location on their Facebook profile. Based on the latest census data from 2011, 53% of the Hungarian population was male and 47% female, furthermore, 41% was under 35 years of age, and 59% was over 35 years of age. Overall, the study population consisted of young people aged 18–35 living in Hungary, with almost the same female-male sex ratio.

### 2.2. Facebook Posts

This study focused on the analysis of Facebook post-level metrics (reach data and interaction data). Therefore, in this subsection, the exclusion criteria and the content classification are shown. It is important to emphasize that only the original content published by the Facebook page was included in the study. Social media content shared by other users was ignored because it may affect the reach data regardless of engagement bait. Since Facebook introduced the sanctioning of engagement bait in Hungarian on 2018, June 25, we analyzed Facebook posts over a three-year period from that date. A total of 1026 social media contents were published between 25 June 2018 and 25 June 2021. Of these, a total of 791 Facebook posts were included in the study based on the exclusion criteria below.

We excluded 99 Facebook posts that did not support quitting (e.g., admin posts, content for ex-smokers and Facebook posts related to passive smoking). Hence, only Facebook posts supporting smoking cessation were included in the study. This allowed us to investigate whether engagement bait content that supports smoking cessation was sanctioned by Facebook or classified as a “benefit to people” and not demoted. Then, we excluded 10 non-image-based social media contents (e.g., Facebook posts containing a video or a link only). This was necessary because algorithmic content ranking is influenced by the type of post (e.g., video or image-based post) [18]. Content ranking is also influenced by the time of publication [18], but there was no need to exclude any content from this point of view, since posts were published at the same time (17:00 on weekdays and 13:00 on weekends). Finally, because content ranking is also influenced by paid advertising, we excluded 126 ‘boosted’ Facebook posts, which were promoted by paid Facebook advertising after publication to reach more users.

In summary, image-based, non-paid Facebook posts which supported smoking cessation and were posted on the same Facebook page at the same time were included. We show some examples of the included content in Appendix A. We also present some excluded content in Appendix A. We classified the included Facebook posts into engagement bait and alternative techniques groups.

“Engagement bait” is a strategy to create Facebook posts that lead people to interact, through likes, shares, comments, and other actions, in order to boost engagement and achieve greater reach in News Feed. The five subcategories for engagement bait were: react baiting, comment baiting, share baiting, tag baiting, and vote baiting. In these cases, the text of the Facebook posts or the used image contained some instructions to encourage users to interact. (e.g., “Like this!” or “Share this!”). The subcategory was named according to the interaction to which it was directed (e.g., “Like this!”—react baiting). These subcategories were designed in accordance with the engagement bait subcategories described in the Facebook Community Standards. The authors’ alternative techniques were applied to ask strategies which encouraged people to interact without engagement bait (without instructions). These strategies use questions rather than instructions. The questions are also aimed at engagement, but indirectly. Categories of alternative techniques were developed according to the subcategories of engagement bait. For example, instead of reacting to baiting, the questions highlighted the emotional background of the Facebook post and focused on the users’ emotions: “This Facebook post illustrates the initial emotions of smoking cessation. What about your feelings? How do you feel in this situation?” The 5 subcategories of the engagement bait group and the alternative techniques group are described below. We present more examples of these subcategories in Appendix A.

Definitions of “engagement bait” subcategories:*React baiting*. Asking people to react to the post (including “Like”, “Love”, “Haha”, “Wow”, “Sad”, and “Angry”).*Comment baiting*. Asking people to comment with specific answers (words, numbers, phrases, or emojis).*Share baiting*. Asking people to share the post with their friends.*Tag baiting*. Asking people to tag their friends.*Vote baiting*. Asking people to vote using reactions, comments, sharing, or other means of representing a vote.

Definitions of “alternative techniques” subcategories:*Questions instead of reaction baiting*. Highlighting the emotional background of the given content. Using questions about emotions related to the given content (e.g., “This Facebook post illustrates the initial emotions of smoking cessation. What about your feelings? How do you feel in this situation?”).*Questions instead of comment baiting*. Using open questions about the topic of the given content (e.g., “We are curious about your experience. What is your experience with this aspect of quitting smoking?”).*Questions instead of share baiting*. Highlighting the benefits of the given content for the community. Using questions that allow Facebook users to identify themselves with the given content (e.g., “This Facebook post helps smoking cessation during pregnancy. Is this aim important to you, too?”).*Questions instead of tag baiting*. Using questions about people who are close to the Facebook user and can benefit from the content (e.g., “Is there anyone in your environment who can draw strength to quit smoking from this Facebook post?”).*Questions instead of vote baiting*. Highlighting the responses of two different communities to the given content. Using questions that allow Facebook users to choose between the highlighted interactions (e.g., “Smokers and non-smokers can express different emotions as a result of this Facebook post. What is your response?”).

### 2.3. Design

The research method of the current study was a hypothesis-generating, retrospective, quantitative content analysis. This is a hypothesis-generating study because it explores a set of data searching for relationships and patterns and then proposes hypotheses that may then be tested in some subsequent study. The current study is retrospective database research where all the events of interest have already happened. We analyzed a Facebook-based smoking cessation intervention using a case-control study method, retrospectively. The content was also classified retrospectively, not prospectively. The participants and the included, classified Facebook posts were presented earlier. In this subsection, we present the definitions of post-level data and the aspects of the analysis. It is important to highlight that these are anonymous and aggregated data, which are subject to Facebook’s privacy policy. Facebook users could not be identified based on these data. Similarly, to page-level data, Facebook makes them available to the page administrators with the users’ consent.

The post-level reach data show how many users have seen the given Facebook post. It is an indicator of algorithmic content ranking. A higher reach value means more users who saw the Facebook post and indicates that the given content is ranked higher in the users’ News Feeds. The opposite is also true. The lower the reach, the fewer users have seen the Facebook post, and the lower Facebook has ranked the given content in the users’ News Feeds. Two types of reach can be distinguished: paid reach and non-paid (organic) reach. Since we excluded advertised Facebook posts from the study, we only used data generated through non-paid reach. Furthermore, Facebook also provides reach data to the page administrators that classify the people reached by the Facebook post: page fans or non-fans. Thus, the users can be grouped based on their previous activity: whether they liked the Facebook page before (page fans) or not (non-fans). The fan reach is the number of people who liked the Facebook page and saw the given Facebook post. The non-fan reach is the number of people who did not like the Facebook page and saw the given Facebook post. The total reach is the sum of fan reach and non-fan reach. Overall, these data show whether Facebook has ranked the content higher or lower in the News Feed of page fans or non-fans.

In general, we calculated the interaction rates by dividing the number of people who used any specific buttons or performed other clicks in relation to the given Facebook post (interactions) by the number of people who saw the post (total reach). The usage of interaction rates is a correction of interaction data. The problem with the absolute number of interactions is that reach data have a direct effect on interaction data. If more Facebook users see the post, they are more likely to use interaction buttons. If the Facebook post gains more interactions, the Facebook algorithm ranks the given content higher in the user’s News Feed [18]. Consequently, we used interaction rates in the current research to express the frequency of the given interaction at the same reach (per one thousand Facebook users). Facebook interactions can indicate how social media content increases the usage of a Facebook-based intervention (reactions, shares, comments, clicks) or decreases it (negative Facebook interactions). We used the total number of negative Facebook interactions during the analysis, because it was available together and not separately. The definitions of the Facebook post-level metrics are described below.

Definitions of “reach data”:*Fan reach*. The number of people who had liked the Facebook page before they saw the given Facebook post.*Non-fan reach*. The number of people who had not liked the Facebook page before they saw the given Facebook post.*Total reach*. The number of people who saw the given Facebook post. The sum of fan reach and non-fan reach.

Definitions of “interaction data”:*Interaction rate*. The number of people who used any specific buttons or performed other clicks in relation to the given Facebook post (interactions) divided by the number of people who saw the post (total reach).*Reaction rate*. The number of people who used a “Like”, “Love”, “Haha”, “Wow”, “Sad” or “Angry” reaction button to express their emotions (reactions) divided by the number of people who saw the post (total reach).*Comment rate*. The number of people who used the ‘comment’ button to publish a text or an image message (comments) divided by the number of people who saw the post (total reach).*Share rate*. The number of people who used the ‘share’ button to send the content to others (shares) divided by the number of people who saw the post (total reach).*Click rate*. The number of people who used any other actions, for example, viewing the Facebook page profile, or expanding photos to full screen (clicks) divided by the number of people who saw the post (total reach).*Engagement rate*. The number of people who used reaction buttons, commented, shared, or clicked on the Facebook post (engagement) divided by the number of people who saw the post (total reach).*The rate of negative interactions*. The number of people who hid the Facebook post, reported the Facebook post as spam or unliked the Facebook page (negative interactions) divided by the number of people who saw the post (total reach).

### 2.4. Procedure

This subsection describes the steps of the analysis. Of the 1026 content posts published between 25 June 2018 and 25 June 2021, 791 Facebook posts (*n* = 791) were included based on the exclusion criteria described earlier. Two raters classified all the 791 Facebook posts separately into engagement bait, alternative techniques, and control group categories (Cohen kappa value of 0.972). Seventy-five Facebook posts met the requirements of the five engagement bait subcategories. In all, 341 contents used alternative techniques without engagement bait. The control group consisted of 375 Facebook posts, which did not use engagement bait or alternative techniques.

The percentage distribution of the five subcategories in the engagement bait group was as follows: react baiting was 16%, comment baiting was 65%, share baiting was 1%, tag baiting was 15% and vote baiting was 3%. Each Facebook post could contain multiple engagement bait subcategories, but only one of each subcategory. The percentage distribution of the five subcategories in the alternative techniques group was as follows: questions instead of reaction baiting (2%), questions instead of comment baiting (94%), questions instead of tag baiting (3%) and questions instead of share/vote baiting (less than 1%). A Facebook post could contain several alternative subcategories, but only one of each subcategory. Due to the low number of items in the subcategories, the engagement bait and alternative technique groups were used together for statistical analysis.

After classifying the posts, statistical analyses were performed. First, we examined the reach data for the study groups and the control group. Next, we compared the rate of different reaction buttons between the groups. Finally, differences in major interaction rates were analyzed. Since the Facebook post-level metrics were not normally distributed, we used a non-parametric statistical test. For statistical analysis, we applied Kruskal-Wallis H test with post hoc Dunn’s test. The effect size was measured by eta squared. All analyses were conducted using the Statistical Package for the Social Sciences software. The *p* value of less than 0.05 was taken to indicate a significant effect, and the *p* value of less than 0.001 was taken to indicate a highly significant effect.

## 3. Results

### 3.1. Algorithmic Content Ranking

In this subsection, we describe how Facebook might restrict the reach of engagement bait and alternative strategy content. If the given Facebook post type reached fewer users on average, it means that the Facebook algorithm has moved that content type further down in the users’ News Feeds. The results are summarized in Table 1. First, the engagement bait group was compared with the control group. Our research question was: “How do engagement bait sanctions work for promoting smoking cessation on Facebook?” We found that fan reach was significantly lower in the engagement bait group compared to the control group. The Kruskal-Wallis H test indicated significance for fan reach (χ^2^(2) = 6.930, *p* = 0.031, η^2^ = 0.006). Dunn’s test was used to identify significant differences. Facebook posts using the engagement bait strategy reached an average of 809.9 people (SD: 428.6; Median: 721), while posts in the control group reached significantly more, an average of 978.1 people (SD: 555.5; Median: 850), based on Dunn’s test (*p* = 0.049). However, no significant differences were observed for non-fan reach and total reach. In summary, the use of the engagement bait strategy did not reduce total reach significantly, it only decreased fan reach.

In the following, the alternative techniques used instead of engagement bait were compared with the control group. Our research question was: “Are there any differences in content ranking or interaction rates, if alternative techniques are used instead of engagement bait?” We found no significant differences between the alternative techniques and the control group for fan reach, non-fan reach, and total reach. It is also worth noting that there was a downward trend for all three accesses. The average reach was the lowest for the engagement bait group (e.g., total reach: 1284.3) and the highest for the control group (e.g., total reach: 1505.7). The average reach of content using the alternative techniques was between the other two groups (e.g., total reach: 1401.5). In summary, although the reach of the alternative techniques was lower compared to the control group, it was not significantly different.

### 3.2. Facebook Reactions

In this subsection, we show how Facebook users used reaction buttons for different content types, which is summarized in Table 2. We first compared the engagement bait group with the control group. Our research question was: “What is the relationship between engagement bait techniques used in Facebook-based smoking cessation intervention and users’ interactions?” We found that significantly fewer “Haha” reaction buttons were used in response to the engagement bait techniques than in the control group. However, no other significant differences were confirmed. The analysis revealed that out of 1000 Facebook users, an average of 11.49 (SD: 5.25) gave a “Like” reaction to engagement bait posts, which is approximately the same as in the control group, where 11.05 (SD: 5.08) used this reaction. Similarly, a nearly identical proportion was observed for the “Sad” reaction. For the three response buttons (“Love”, Angry”, and “Wow”), the interaction rate was higher in the engagement bait group compared to the control group, but the differences were not significant. Using the Kruskal-Wallis H test, we observed a significant correlation only in the “Haha” reaction rate (χ^2^(2) = 15.818, *p* < 0.001, η^2^ = 0.018). Dunn’s pairwise test confirmed the difference between the engagement bait techniques and the control group. Out of 1000 Facebook users, an average of 1.38 people (SD: 2.91) reacted “Haha” to content, which is significantly more than in the control group, where 2.37 people (SD: 3.71) used this reaction. Overall, a significant difference was only confirmed for the “Haha” response, which was higher in the control group compared to the engagement bait techniques.

Secondly, the alternative techniques were compared with the control group. Our research question was: “Are there any differences in content ranking or interaction rates, if alternative techniques are used instead of engagement bait?” We observed that even for the alternative techniques, there was a significant difference only for the “Haha” reactions. For the reactions “Like”, “Love”, “Wow” and “Sad”, the average interaction rate was almost the same in the alternative techniques and the control group. The “Angry” reaction rate was higher in the alternative techniques group, but not significantly. As previously shown, the Kruskal-Wallis H test indicated a significant difference in the “Haha” response (χ^2^(2) = 15.818, *p* < 0.001, η^2^ = 0.018). The post hoc Dunn’s pairwise test revealed a significant difference between the alternative techniques and the control group. Out of 1000 Facebook users, an average of 1.52 (SD: 2.97) gave a “Haha” reaction to posts using alternative techniques, which was significantly more than in the control group, where 2.37 (SD: 3.71) used this reaction. In summary, the control group had significantly more “Haha” reactions compared to the alternative techniques group.

### 3.3. Facebook Users’ Interactions

In this subsection, we describe what interaction buttons Facebook users applied for the different content types, which is summarized in Table 3. Firstly, the engagement bait group was compared with the control group. Our research question was: “What is the relationship between engagement bait techniques used in Facebook-based smoking cessation intervention and users’ interactions?” No significant difference was observed between the engagement bait group and the control group for either interaction rate. Notably, the reaction rate, comment rate, share rate, click rate and engagement rate were slightly higher for the engagement bait group compared to the control group. However, the rate of negative interactions was similar in the two groups. In summary, we did not observe any significant difference between the engagement bait group and the control group, which may be due to the artificial back-ranking of engagement bait content.

Secondly, the alternative techniques were compared with the control group. Our research question was: “Are there any differences in Facebook sanctions or users’ interactions, if alternative techniques are used instead of engagement bait during a smoking cessation intervention?” We found that the alternative techniques had a significantly lower reaction rate and rate of negative interactions, and a significantly higher click rate compared to the control group. No significant difference was found for the other interaction rates. Using the Kruskal-Wallis H test, significant correlations were observed for reaction rate (χ^2^(2) = 10.492, *p* = 0.005, η^2^ = 0.011), rate of negative interactions (χ^2^(2) = 6.891, *p* = 0.032, η^2^ = 0.006), and click rate (χ^2^(2) = 8.072, *p* = 0.018, η^2^ = 0.008). Dunn’s pairwise test demonstrated a significant difference between the alternative techniques and the control group for all three interaction rates. Out of 1000 Facebook users, an average of 13.02 (SD: 7.81) used a reaction button for posts with the alternative strategy, and 0.06 (SD: 0.20) produced a negative Facebook interaction. These values are significantly lower compared to the control group, where an average of 14.46 people (SD: 7.76) utilized a reaction button (*p* = 0.006), and 0.12 people (SD: 0.34) used a negative Facebook interaction (*p* = 0.028). The click rate was significantly higher in the alternative techniques group (*p* = 0.021). Out of 1000 Facebook users, posts using alternative techniques received an average of 46.16 clicks (SD: 41.48), while the control content had 37.30 clicks (31.98). There was no difference in the share rate between the two groups. The comment rate and engagement rate were slightly higher for the alternative strategies, but this difference was not significant. To sum up, the disadvantage of using alternative strategies may be that they can reduce the reaction rate, but they have the dual advantage of increasing the click rate while reducing the rate of negative interactions.

## 4. Discussion

### 4.1. Principal Results

The primary aim of this research was to determine the impact of engagement bait on content ranking and Facebook user activity. Our results suggest that Facebook ranks engagement bait content lower in the page fans’ News Feed. Fan reach was significantly lower in the engagement bait group compared to the control group. This result provides an important insight into the mechanism of sanctioning. The page fan group offers Facebook a better opportunity for sanctioning because the page fan group can be defined better than the non-fan group. At the same time, previous research has shown that a high proportion of “page like” activities can be associated with a high engagement rate [19]. This suggests that restricting fan reach offers a more vulnerable point for Facebook to sanction engagement bait. It is assumed that engagement rate or other interaction rates can be better sanctioned indirectly by restricting fan reach. In summary, Facebook sanctions engagement bait content among page fans. Our results show that Facebook did not exempt smoking cessation support content from sanctioning. Thus, Facebook did not classify them as public health content and a “benefit to people”, even though they are considered socially useful.

The relationship between engagement bait techniques and Facebook users’ activity was also explored. Significantly fewer “Haha” reaction buttons were used in response to the engagement bait techniques than in the control group. This may be due to the restriction of fan reach. Our previous research revealed that the “Haha” reaction button is a fan-specific interaction [18]. Furthermore, of all Facebook interactions, the “Haha” reaction button correlated most strongly with fan reach [18]. Therefore, this reaction button may be the most sensitive to fan reach restriction. Artificially reducing the fan reach of engagement bait content was associated with a reduction in the application of the “Haha” reaction button. This phenomenon highlights that restricting fan reach may lead to a decrease in interaction rates. Without sanctions, we would expect engagement bait techniques to increase the engagement rate. Contrary to expectations, in no case was the engagement rate or other interaction rate significantly higher in the engagement bait group compared to the control group. To sum up, it seems that by artificially reducing fan reach, Facebook can ensure that interaction rates are not significantly different between the engagement bait group and the control group.

It is also worth noting that there was no difference in the rate of negative interactions between the engagement bait and the control group. This is surprising because Facebook sanctions engagement bait content, since users do not enjoy such Facebook posts. This is contradicted by the result that the rate of negative interactions was not lower for the engagement bait. Furthermore, this association should be considered in the context of a significantly lower value of fan reach for the engagement bait group. Although Facebook ranked engagement bait content lower in the News Feed of page fans, there was still no difference in the rate of negative interactions. Thus, it is mainly true for non-fan users that they did not use negative Facebook interaction buttons more often in response to engagement bait content than to content without engagement bait.

The secondary aim was to find alternative techniques instead of engagement bait that can stimulate engagement without obtaining a lower ranking in the user’s News Feed. No significant differences were found between the alternative techniques and the control group for fan reach, non-fan reach, and total reach. This could imply that Facebook did not consider this content as engagement bait and thus did not rank it lower in the users’ News Feed. In other words, alternative strategies seem to be able to circumvent Facebook’s engagement bait restrictions in terms of organic reach. Furthermore, we found that the alternative techniques had significantly lower rates of negative interactions and significantly higher click rates compared to the control group. Reducing the rate of negative interactions (resistance to behavior change) can be a remarkable advantage, especially in the field of addiction [22,30]. Since all alternative techniques ask questions, the lower rate of negative interactions can easily be attributed to the use of questions rather than instructions. This may be an advantage when using questioning strategies for smoking cessation interventions on Facebook. On the other hand, increasing the click rate can be beneficial in public health campaigns in general [28,32].

However, a significant decrease in reaction rate was observed in the alternative strategies group compared to the control group. This is presumably due to a reduction in the rate of “Haha” reactions, as no significant differences were found between the two groups for the other reaction buttons. There are two possible explanations for this phenomenon. One of them is that Facebook may classify alternative strategies as a kind of intermediate group between engagement bait content and content without engagement bait. Therefore, Facebook may artificially reduce the reach of alternative content to an insignificant extent only. As mentioned earlier, since the “Haha” reaction button is the most sensitive to the fan reach constraint, the restrictions by Facebook may have reduced the “Haha” rate and consequently may have reduced the reaction rate as well. Thus, the significant differences in reaction button performance may have been caused by the mild sanctioning of fan reach. Another explanation is that the composition of the alternative strategies may be responsible for the low reaction rate. Most alternative strategies were targeted at comments, not reaction buttons. Furthermore, it is possible that the users responded to these questioning strategies by commenting rather than using reaction buttons. In conclusion, further studies are needed to explore whether the alternative strategies are suitable for stimulating reaction button use.

### 4.2. Hypotheses for Future Research

This study was an exploratory, hypothesis-generating research to gain insights into the mechanism of engagement bait on Facebook and its alternatives. We present some recommended hypotheses for more rigorous testing in the future. First, the hypotheses related to engagement bait are discussed. Our results assume that engagement bait content tends to be demoted in the page fans’ New Feed. This suggests that further investigation of the mechanism of engagement bait will be an exciting area of research. It is also conceivable that Facebook ranks the engagement bait content lower for more active users. Facebook can achieve an even greater reduction in interaction rate by restricting the reach of page fans who frequently use the interaction buttons. However, we were not able to investigate this aspect, because we used aggregated and anonymized data. However, future prospective research can answer this research question by recording who the interaction is from (e.g., fans using a reaction button or fans making a comment) and involving page fans in the study who add a pre-arranged interaction to each post.

Furthermore, our results highlight that the engagement bait content is less likely to increase the engagement rate or each interaction rate (reaction rate, comment rate, share rate or click rate) due to the reduced fan reach. It is therefore recommended that future research consider the level of constraints that may be imposed on these data. Presumably, the engagement rate and other interaction rates are the endpoints of sanctioning the engagement bait. Thus, Facebook reduces fan reach to the extent that there is no significant difference in interaction rates between engagement bait content and content without engagement bait. If more similar studies are conducted in the future, the extent to which fan reach is reduced could be compared. This will allow the level of the sanctions on the interaction rates to be determined. As a first step, this could be explored through retrospective research. In a second step, it can be tested by prospective studies, which would help to better understand sanctions.

The results of the present study suggest that Facebook users are not likely to use more negative interaction buttons related to engagement bait content than content without engagement bait. This hypothesis may identify an important area of research that questions the legitimacy of engagement bait. In the future, Facebook users’ attitudes toward engagement bait content could be assessed through questionnaires. Our research has shown that Facebook users’ attitudes toward such content are not necessarily negative. It should be highlighted that content labeled as engagement bait by Facebook is essentially interaction oriented. The effectiveness of interactive content in public health interventions has been demonstrated in several studies [32]. For instance, the perception of “vote bait” content can be positive if the vote is not created for commercial purposes but to express opinions on relevant social issues.

Hypotheses for future testing regarding engagement bait:Engagement bait content tends to be demoted in the page fans’ New Feed.Due to the reduced fan reach, engagement bait content cannot increase the engagement rate or each interaction rate (reaction rate, comment rate, share rate or click rate).Facebook users do not use more negative interaction buttons related to engagement bait content than content without engagement bait.

Second, the hypotheses related to alternative techniques are discussed. Based on our results, we assume that alternative techniques would stimulate the click rate and inhibit the rate of negative interactions without significantly reducing organic reach. A high click rate could be relevant for those planning Facebook-based public health interventions. A low rate of negative interactions is particularly relevant for interventions where resistance from participants (e.g., individuals with addiction) can be expected. In the future, this hypothesis should be tested in randomized controlled trials. Randomization was not possible in our retrospective study. However, in future prospective research, the alternative techniques group could be determined by randomly assigning specific questions as experimental units to the same intervention content (e.g., “What is your opinion?”). Furthermore, the engagement bait group could be identified by allocating instructions as experimental units for the same intervention content (e.g., “Comment “YES”!”). These randomized, longitudinal studies would be suitable to reveal the difference between engagement bait and alternative techniques, without distortion due to different post designs.

Our results also suggest that asking strategies that encourage people to interact may result in fewer Facebook reactions, especially the “Haha” reaction. This hypothesis could be properly tested by examining the subcategories. For future prospective research, it is recommended to compare the different subcategories with each interaction rate. For example, it might be exciting to explore how “react baiting” and “ questions instead of react baiting” are related to the reaction rate. In our current research, the proportion of “comment baiting” and “questions instead of comment baiting” subcategories was rather high, and the proportion of other subcategories was low. This did not allow for a comparison of each subcategory with its corresponding interaction rate. However, examining the subcategories would highlight the potential flaws of alternative techniques. Thus, asking strategies could be improved, or new asking strategies could be developed in the future.

Hypotheses for future testing regarding alternative techniques:Alternative techniques can stimulate the click rate and inhibit the rate of negative interactions without significantly reducing organic reach.Asking strategies that encourage people to interact may result in fewer Facebook reactions, especially “Haha” reaction.

### 4.3. Limitations and Strengths

Finally, some limitations of this research need to be considered. Due to the retrospective nature of the study, there is a disproportionality in the number of items in the subcategories. Engagement bait and the use of alternative techniques were not considered in the preparation of the intervention content. We classified the content into these groups or subcategories subsequently. This disproportionality was corrected by summing up the subcategories. Furthermore, Facebook’s Community Standards identify repeated use of engagement bait as a factor that increases sanctioning. Facebook pages that repeatedly and systematically use engagement bait are more likely to be demoted than individual posts. The Facebook page under investigation also repeatedly posted engagement bait content during the study period; therefore, the results should be interpreted in this context. Another limitation may be the timing of the Facebook posts. In this study, all content appeared regularly, on different days, but at the same time. Therefore, the daily timing was the same between the study and control groups. A previous study shows that the organic reach of social media content gradually increased from one year to another, and the Facebook algorithm of content ranking took the interactions into account to varying degrees for the calculation of total organic reach [18]. During the current retrospective study, there was no disproportion in the dates of the investigated Facebook posts. Hence, the annual timing may have the same effect on the study and control groups.

One of the strengths is the three-year study period, which allowed us to select 791 Facebook posts out of 1026 social media contents. As we have published content almost daily since the “engagement bait” regulation in 2018, this database size seems to be notable. Another strength of the research is that the results are generalizable to any public health intervention. The algorithmic content ranking is independent of the public health focus. The order of the content in a Facebook user’s News Feed does not depend on the topic of the Facebook post, whether it features smoking cessation support or cardiovascular prevention. The content ranking is determined by other factors, such as the timing or the performance of the posts [18,20]. Furthermore, content ranking is independent of geographic or language barriers. Although the content was in Hungarian, Facebook used the same algorithm it applied for posts in English or any other language. Our results on engagement bait content and the alternatives are therefore widely applicable.

## 5. Conclusions

This is the first study to investigate the sanctioning of engagement bait in the context of a Facebook-based public health intervention. We are the first to succeed in exploring how Facebook ranks engagement bait content backward in the user’s News Feeds. The fan reach on the post level could be the intervention point where Facebook’s algorithm performs the restriction. As a result of this sanctioning, engagement bait content is not able to increase the engagement rate or any interaction rate. In summary, engagement bait content is not recommended for Facebook-based public health interventions.

Furthermore, there are some exciting conclusions to be drawn from the results on alternatives to engagement bait. These asking strategies can help health professionals to avoid sanctioning engagement bait in Facebook-based public health interventions. Alternative techniques can stimulate the click rate and inhibit the rate of negative interactions without significantly reducing organic reach. Increasing the click rate is also important because Facebook also sanctions “click bait” content. Therefore, alternative techniques could presumably avoid sanctioning “click bait”. In addition, increasing the click rate in public health campaigns in general can be beneficial [28,33]. Further prospective research is needed to test and develop alternative strategies.

Finally, it is striking that there was no difference in the rate of negative interactions between the engagement bait group and the control group. This raised concerns regarding the legitimacy of sanctioning engagement bait content. This important research focus requires further investigation. Furthermore, our research also shows that Facebook did not exempt content that supports smoking cessation from sanctioning engagement bait. However, other socially relevant content (e.g., missing child report) is categorized by Facebook as a “benefit to people” and they are not sanctioned. It would be useful to have a consultation between Facebook and international public health organizations to exempt public health content from Facebook sanctions, such as restrictions on engagement bait, due to their social benefits. Although the results refer to Facebook, the methodology of the current research may be useful for investigating sanctions on other social media platforms, including WeChat and Twitter.

## Figures and Tables

**Table 1 ijerph-20-00958-t001:** The median of fan, non-fan, and total organic reach of engagement bait, alternative techniques, and the control group.

	Organic Reach, Median		
	Engagement Bait	Alternative Techniques	Control Group
Fan Reach	721 ^a^	814	850 ^a^
Non-fan Reach	384	371	375
Total Reach	1177	1238	1266

^a^ Significant difference, *p* < 0.05 (2-tailed). ^b^ Highly significant difference, *p* < 0.001 (2-tailed).

**Table 2 ijerph-20-00958-t002:** The mean and the SD of interaction rate by engagement bait, alternative techniques, and the control group.

	Interaction Rate, Mean (SD)		
	Engagement Bait	Alternative Techniques	Control Group
“Like” Rate	11.49 (5.25)	10.39 (5.22)	11.05 (5.08)
“Love” Rate	1.34 (3.74)	0.76 (3.03)	0.71 (3.05)
“Haha” Rate	1.38 ^b^ (2.91)	1.52 ^b^ (2.97)	2.37 ^b^ (3.71)
“Wow” Rate	0.92 (3.52)	0.14 (0.74)	0.20 (1.33)
“Sad” Rate	0.15 (0.47)	0.09 (0.39)	0.11 (0.61)
“Angry” Rate	0.04 (0.16)	0.11 (0.75)	0.02 (0.17)

^a^ Significant difference, *p* < 0.05 (2-tailed). ^b^ Highly significant difference, *p* < 0.001 (2-tailed).

**Table 3 ijerph-20-00958-t003:** The mean and the SD of interaction rate by engagement bait, alternative techniques and the control group.

	Interaction Rate, Mean (SD)		
	Engagement Bait	Alternative Techniques	Control Group
Reaction Rate	15.32 (10.10)	13.02 ^a^ (7.81)	14.46 ^a^ (7.76)
Comment Rate	2.10 (3.58)	2.19 (3.73)	1.64 (2.84)
Share Rate	1.34 (1.38)	1.25 (1.19)	1.27 (1.20)
Click Rate	45.24 (36.22)	46.16 ^a^ (41.48)	37.30 ^a^ (31.98)
Engagement Rate	79.31 (39.78)	75.63 (46.72)	69.13 (38.05)
The Rate of Negative Interactions	0.13 (0.43)	0.06 ^a^ (0.20)	0.12 ^a^ (0.34)

^a^ Significant difference, *p* < 0.05 (2-tailed). ^b^ Highly significant difference, *p* < 0.001 (2-tailed).

## Data Availability

Original data in Appendix A. The data sets used and analyzed during the current research are available from the corresponding author on reasonable request.

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
