# Peer review of "How to Avoid Lower Priority for Smoking Cessation Support Content on Facebook: An Analysis of Engagement Bait"

_ijerph, 2023, doi:10.3390/ijerph20020958_

Round 1

Reviewer 1 Report

This is a novel and compelling manuscript that explore the impact of engagement bait and alternative techniques o Facebook users from the perspective of public health. I have minor suggestions as follows: 

 1. The timing might influence the results. If the post numbers of engagement bait, alternative techniques, and control group were totally different within the three years, the total organic reach and other results might be influenced. How did authors control the bias? Authors need to provide further details.

2. If the results (such as fan reach) were not normally distributed, I suggest authors use the median value instead of the mean.

 3. In the discussion and conclusion sections, authors could provide more suggestions about public health interventions and the sanctions not just for Facebook, but also for other social media including WeChat and Twitter et al. 

Author Response

Response to Reviewer 1 Comments

We wish to thank you all for your constructive comments. Your comments provided valuable insights to refine its contents. We try to address the issues raised as best as possible.

Reviewer 1. / Comment 1.

This is a novel and compelling manuscript that explore the impact of engagement bait and alternative techniques o Facebook users from the perspective of public health. I have minor suggestions as follows: 1. The timing might influence the results. If the post numbers of engagement bait, alternative techniques, and control group were totally different within the three years, the total organic reach and other results might be influenced. How did authors control the bias? Authors need to provide further details.

RESPONSE: Thank you to the reviewer for pointing this out. In response to this comment, the following sentence has been added to the “Limitations” section (in Discussion): “Another limitation may be the timing of the Facebook posts. In this study, all contents appeared regularly, on different days, but at the same time. Therefore, the daily timing was the same between the study and control groups. A previous study shows that the organic reach of social media contents gradually increased from one year to another, and the Facebook algorithm of content ranking took the interactions into account to varying degrees for the calculation of total organic reach [15]. During the current retrospective study, there was not a disproportion in the dates of the investigated Facebook posts. Hence, the annual timing may have the same effect on the study and control groups.” We hope you will find it correct.

Reviewer 1. / Comment 2.

  1. If the results (such as fan reach) were not normally distributed, I suggest authors use the median value instead of the mean.

RESPONSE: Thank you so much for your comment. In response to this comment, we used the median value of organic reach. However, the medians of interaction rates and engagement rates were zero in many cases. That is why, we preferred using the "mean" in these cases. Hopefully, you will find this response satisfactory.

Reviewer 1. / Comment 3.

  1. In the discussion and conclusion sections, authors could provide more suggestions about public health interventions and the sanctions not just for Facebook, but also for other social media including WeChat and Twitter et al.

RESPONSE: Thank you for the valuable observation. In response to this comment, the following sentences have been added to the " Although the results refer to the Facebook, the methodology of the current research may be useful for the investigating sanctions on other social media platforms, including WeChat or Twitter." Hopefully, now you will find it acceptable.

We thank you for the time you put in reviewing our paper and look forward to meeting your expectations. We hope that this new version is satisfactory.

Reviewer 2 Report

Dear Authors,

Thank you for the opportunity to review an interesting article entitled: How to avoid lower priority for smoking cessation support contents on Facebook: an analysis of engagement bait'. The aim of this article is to determine the negative effect of engagement bait and find alternative techniques. This topics is of a great interest to both social media industry and public health professionals. Overall this article is well written and executed, and will make a contribution to the literature that can be used and cited by future researchers. However, the following comments, which aim to improve the text, are worth considering:

Comment 1: Line 109: The data in this paragraph should be compared with census data to demonstrate representativeness.

Comment 2: Line 178: Alternative techniques are all asking questions. Some more different categories are expected.

Comment 3: Line 283: 94% of the percentage distribution of the five subcategories in the alternative techniques group were questions instead of comment baiting. The information indicated so-called alternative techniques were basically only one approach.

Comment 4: Line 296: Please standardize the notation of the "p" coefficient - once there is "P=..." and once there is "p-value...".

Comment 5: Line 383: Alternative techniques had a significantly lower rate of negative interactions. According Line 168, the rate of negative interactions are defined as the number of people reported the Facebook post as a spam, or unliked the Facebook page. Again, since alternative techniques are all asking questions, lower rate of negative interactions can be easily inferred, because asking questions generally are not considered as a spam or unliked post.

Author Response

Response to Reviewer 2 Comments

We wish to thank you all for your constructive comments. Your comments provided valuable insights to refine its contents. We try to address the issues raised as best as possible.

Reviewer 2. / Comment 1.

Thank you for the opportunity to review an interesting article entitled: How to avoid lower priority for smoking cessation support contents on Facebook: an analysis of engagement bait'. The aim of this article is to determine the negative effect of engagement bait and find alternative techniques. This topic is of a great interest to both social media industry and public health professionals. Overall, this article is well written and executed, and will make a contribution to the literature that can be used and cited by future researchers. However, the following comments, which aim to improve the text, are worth considering: Comment 1: Line 109: The data in this paragraph should be compared with census data to demonstrate representativeness.

RESPONSE: Thank you for your valuable observation. In response to your comment, the following sentences have been added to the "Participants" section (in Methods): “Based on the latest census data from 2011, 53% of the Hungarian population was male and 47% female, furthermore, 41% was under 35 years of age, and 59% was over 35 years of age.” Hopefully, you will find it satisfactory.

Reviewer 2. / Comment 2.

Comment 2: Line 178: Alternative techniques are all asking questions. Some more different categories are expected.

RESPONSE: Thank you so much for your comment. In response to this comment, we have modified the manuscript in the Methods section as follows: “The questions are also aimed at engagement, but indirectly. Categories of alternative techniques were developed according to the subcategories of engagement bait.” Hopefully, you will find it correct.

Reviewer 2. / Comment 3.

Comment 3: Line 283: 94% of the percentage distribution of the five subcategories in the alternative techniques group were questions instead of comment baiting. The information indicated so-called alternative techniques were basically only one approach.

RESPONSE: Thank you for pointing these out. We absolutely agree with the reviewer. That is why, we clarify this problem in the "Limitation" section: "Due to the retrospective nature of the study, there is a disproportionality in the number of items in the subcategories. Engagement bait or the use of alternative techniques were not considered in the preparation of the intervention content. We classified the content into these groups or subcategories subsequently. This disproportionality was corrected by summing up the subcategories." We also refer to this bias in the "Hypotheses for future research" section: "In the future, this hypothesis should be tested in randomized controlled trials. Randomization was not possible in our retrospective study. However, in future prospective research, the alternative techniques group could be determined by randomly assigning specific questions as experimental units to the same intervention contents (e.g., "What is your opinion?"). Furthermore, the engagement bait group could be identified allocating instructions as experimental units for the same intervention contents (e.g., "Comment “YES”!"). These randomized, longitudinal studies would be suitable to reveal the difference between engagement bait and alternative techniques, without distortion due to different post design." In the current exploratory study, the 2 big group (engagement bait and alternative techniques) were compared. The associations of 5 and 5 subcategories may be investigated in the future. Hopefully, you will find it satisfactory.

Reviewer 2. / Comment 4.

Comment 4: Line 296: Please standardize the notation of the "p" coefficient - once there is "P=..." and once there is "p-value...".

RESPONSE: Thank you for the observation. We standardized the notation of the "P" coefficient. Hopefully, you will find it correct.

Reviewer 2. / Comment 5.

Comment 5: Line 383: Alternative techniques had a significantly lower rate of negative interactions. According Line 168, the rate of negative interactions are defined as the number of people reported the Facebook post as a spam, or unliked the Facebook page. Again, since alternative techniques are all asking questions, lower rate of negative interactions can be easily inferred, because asking questions generally are not considered as a spam or unliked post.

RESPONSE: Thank you very much for your comment. In response to this comment these sentences was added to the "Discussion" section: "Since all alternative techniques ask questions, the lower rate of negative interactions can easily be attributed to the use of questions rather than instructions. This may be an advantage when using questioning strategies in smoking cessation interventions on Facebook." Hopefully, you will find this response acceptable.

We thank you for the time you put in reviewing our paper and look forward to meeting your expectations. We hope that this new version is satisfactory.

Reviewer 3 Report

This manuscript aimed to explore the impact of engagement bait on content ranking and the Facebook user activity based on a Hungarian smoking prevention Facebook page. I think that the topic of this manuscript does not meet with the scope of IJERPH. Moreover, both the study and the manuscript has several serious weaknesses. The title is misleading as there is nothing about smoking cessation in the manuscript. Major weaknesses are listed below.

The abstract does not reflect clearly the content of the main text.

In the Introduction, there is nothing about smoking cessation support in social media. It would be essential to discuss briefly the literature background of this topic. Moreover, it would be valuable to include some information about tobacco and nicotine use in Hungary and why cessation support on social media would be relevant in this country.

It seems that at many times, the Authors incorrectly use self-citation, especially Ref. No. 15.

In line 47, ’engagement on Facebook’ is a general term, but the Authors put it into to context of public health without any introduction.

Research aims do not consider smoking cessation support on Facebook. Aims are just generally related to Facebook engagement bait. It would be essential to specify study aims and to provide clear and relevant aims.

Methods section is the weakest point of this manuscript. In subsection 2.1., the Authors mention an ’intervention’, however, it is unclear why an intervention it is. Moreover, what type of intervention and what is the structure of the intervention?

In lines 104-107, various smoking prevention approaches were presented. However, in the manuscript, the evidence behind applying these approaches is unclear. Besides, citations seem to be irrelevant self-citations. Please provide correct citations supporting the applied methods. Furthermore, consider that this manuscript targets smoking cessation instead of non-smokers and smokers unwilling to quit completely.

Regarding ’gender’, please consider SAGER guideline.

Lines 167-196 are unnecessary paragraphs. These could be easily included in the text. Alternatively, the Authors should prepare a table combining and presenting engagement bait and alternative techniques whit examples.

Lines 202-203. Please clarify the study design. Please revise the study design whether it is really a hypothesis-generating study or not. Please provide citation for this study design.

Repetition of previously described things commonly occur in the text. This should be avoided.

Lines 225-239. Too long description for interaction rate. It does not fit to the methods section of a scientific manuscript. Please write concise and clear sentences to present measures of the current study.

Lines 242- 270. Bullet points in this part are unnecessary repetitions of previously described measures. There is not any citation for applying these measures.

Procedure of Methods section usually discusses the way of data collection. Statistical analysis/methods is a separate subsection of Methods.

It is unclear what the Authors regarded as ’control group’.

It is unclear how post classification was conducted. Do you mean classifying posts into engagement bait vs alternative techniqus vs control? If yes, why not provide a clear description for classification and allocation into study groups?

Line 298. ’…the p valuse of less than 0.001 was taken to indicate a highly significant effect.’ Why not calculated effect sizes? It is an incorrect consideration of significance.

The interpretation of the Hungarian language by Facebook was not mentioned in Methods.

It seems that this Facebook page has a small reach of users. Moreover, ignoring several other factors connected to the operation of the algorithm, it is impossible to draw reliable conclusions in this study.

In Results section, some of the research questions did not meet with the applied statistical methods. E.g., in lines 306-314, a more specific research question would be needed. Here, the Authors explored possible differences between fan group reach vs non-fan group reach of posts categorized in the engagement bait and control groups. Furthermore, presented statistical tests do not support this statement. Without presenting clearly your findings e.g., in a table, these results are questionable.

In line 337, the Authors mentioned ’1,000 Facebook users. The exact sample size is unclear. Generally, presenting clearly descriptive characteristics of this study is missing. E.g., presenting in a table the n of engagement bait posts, alternative posts, control posts, n of fan vs non-fan groups, etc. in light of some demographic variables.

Some research questions are repeated several times which indicate inadequate approach of study aims.

Discussion and especially Conclusions are not scientifically sound.

Author Response

Response to Reviewer 3 Comments

We wish to thank you all for your constructive comments. Your comments provided valuable insights to refine its contents. We try to address the issues raised as best as possible.

Reviewer 3. / Comment 1.

This manuscript aimed to explore the impact of engagement bait on content ranking and the Facebook user activity based on a Hungarian smoking prevention Facebook page. I think that the topic of this manuscript does not meet with the scope of IJERPH. Moreover, both the study and the manuscript have several serious weaknesses. The title is misleading as there is nothing about smoking cessation in the manuscript. Major weaknesses are listed below.

RESPONSE: Thank you to the reviewer for pointing this out. In response to this comment, the following sentence has been added to the “Methods” section: “The goal of the intervention is to support smoking cessation, and not to prevent smoking. A previous cross-sectional study showed that this Facebook page had a significantly positive effect on smoking habits, knowledge, and attitudes about smoking cessation [10]. A third of ex-smoker participants reported that this Facebook page helped them quit smoking [10]. In addition, two-thirds of the ex-smoker participants stated that following the Facebook page helped prevent relapse [10].” We hope you will find this response correct.

Reviewer 3. / Comment 2.

The abstract does not reflect clearly the content of the main text.

RESPONSE: Thank you for this suggestion. We reviewed the abstract and found no inaccuracies. If you write a specific instruction, we will correct it immediately. Hopefully, you will find this response acceptable.

Reviewer 3. / Comment 3.

In the Introduction, there is nothing about smoking cessation support in social media. It would be essential to discuss briefly the literature background of this topic. Moreover, it would be valuable to include some information about tobacco and nicotine use in Hungary and why cessation support on social media would be relevant in this country.

RESPONSE: Thank you for your valuable observation. In response to this comment, the following sentence has been added to the “Introduction” section: “Facebook platform has many advantages in supporting smoking cessation. For example, it can be a useful tool to contact hard-to-reach smokers [8]; it can be more cost-effective than television advertising [9]; or it can effectively help young smokers to quit [10]. These benefits of Facebook-based smoking cessation interventions can mean even more in a country with high smoking rates, like Hungary, where the proportion of smokers in the Hungarian adult population is 28.7% [11].” Hopefully, you will find it satisfactory.

Reviewer 3. / Comment 4.

It seems that at many times, the Authors incorrectly use self-citation, especially Ref. No. 15.

RESPONSE: We thank the reviewer for this observation. A total of four self-citation out of thirty references (less than 15%). Each self-citation provides additional information about the investigated Facebook page. Two self-citations are focused on the "smoking cessation method" of the Facebook page (the "motivational interviewing" approach). One self-citation needs to present the "smoking cessation results" of the Facebook page. Lastly, one self-citation (Ref. No. 15.) is very closely to the current research, because it investigated the content ranking ("how the Facebook Algorithm works") in this Facebook-based smoking cessation intervention. As we know after literature review, there is no other reference that comes so close to the current research goal. That is why, we referred to this self-citation (Ref. No. 15.) more than 15 times in the manuscript. Hopefully, you will find this response acceptable.

Reviewer 3. / Comment 5.

In line 47, ’engagement on Facebook’ is a general term, but the Authors put it into to context of public health without any introduction.

RESPONSE: Thank you so much for your comment. In response to this comment, the following sentence has been added to the "Introduction" section: "Engagement" is a widely used concept in web-based public health interventions. Hopefully, you will find this response satisfactory.

Reviewer 3. / Comment 6.

Research aims do not consider smoking cessation support on Facebook. Aims are just generally related to Facebook engagement bait. It would be essential to specify study aims and to provide clear and relevant aims.

RESPONSE: We thank to the reviewer for pointing this out. In response to this comment, research aims were modified in the "Introduction" section: “How do engagement bait sanctions work for smoking cessation support contents on Facebook?” “What is the relationship between engagement bait techniques in smoking cessation support contents and Facebook users' activity, especially interaction rates?” “Are there any differences in sanctions or interaction rates, if alternative techniques are used instead of engagement bait in smoking cessation support contents?” Hopefully, you will find it correct.

Reviewer 3. / Comment 7.

Methods section is the weakest point of this manuscript. In subsection 2.1., the Authors mention an ’intervention’, however, it is unclear why an intervention it is. Moreover, what type of intervention and what is the structure of the intervention?

RESPONSE: Thank you for your comment. This is a web-based smoking cessation intervention. The type of intervention is “Facebook-based”. Hopefully, you will find it satisfactory.  

Reviewer 3. / Comment 8.

In lines 104-107, various smoking prevention approaches were presented. However, in the manuscript, the evidence behind applying these approaches is unclear. Besides, citations seem to be irrelevant self-citations. Please provide correct citations supporting the applied methods.

RESPONSE: Thank you for your suggestion. Firstly, it is widely recognized that MI (motivational interviewing) is an evidence-based psychological method to support smoking cessation. All major smoking cessation guidelines (e.g., US, UK, EU, AU) recommend the motivational interviewing method. However, the motivational interviewing is not the focus of the current research. The motivational interviewing on Facebook is a new area of research. The most relevant reference in this topic is the "How to create social media contents based on Motivational Interviewing approach to support tobacco use cessation?" article, because this article investigated the smoking cessation intervention of the current research. By reading this article, the Reviewer will understand how this smoking cessation intervention works. Secondly, "smoking prevention" and "smoking cessation" are not the same. We noticed that the Reviewer mixes these two concepts. "Smoking prevention" is a way to avoid smoking, while "smoking cessation" is a way to stop smoking. We never used the term of "smoking prevention" in the manuscript. On the other hand, we added this sentence in the Method section to clarify the misunderstanding: "The goal of the intervention is to support smoking cessation, and not to prevent smoking". Hopefully, now you will find it acceptable.  

Reviewer 3. / Comment 9.

Furthermore, consider that this manuscript targets smoking cessation instead of non-smokers and smokers unwilling to quit completely.

RESPONSE: Thank you for your comment. We presented the smoking cessation intervention and its goals in the Method section. This is a public Facebook page which means that every Facebook user (smokers, non-smokers, ex-smokers) can see the Facebook posts. On the other hand, we used only smoking cessation support contents for the analysis. The Reviewer can follow this analysis in the Method section. Hopefully, you will find it satisfactory.

Reviewer 3. / Comment 10.

Regarding ’gender’, please consider SAGER guideline.

RESPONSE: We agree with the reviewer. The term “gender” was changed to “sex” in the manuscript. Hopefully, you will find it correct.

Reviewer 3. / Comment 11.

Lines 167-196 are unnecessary paragraphs. These could be easily included in the text. Alternatively, the Authors should prepare a table combining and presenting engagement bait and alternative techniques whit examples.

RESPONSE: Thank you for this observation. The description of subcategories is an important part for clarity. We would like to follow the IJERPH guidelines for authors, which offers a list in this case. We also thought about using the table, but since IJERPH's style is the list, we did not want to deviate from it.  However, we present more examples of these subcategories in Supplementary Materials File S1. Hopefully, now you will find it satisfactory.  

Reviewer 3. / Comment 12.

Lines 202-203. Please clarify the study design. Please revise the study design whether it is really a hypothesis-generating study or not. Please provide citation for this study design.

RESPONSE: Thank you for pointing these out. This is really a hypothesis-generating study, because it explores a set of data searching for relationships and patterns, and then proposes hypotheses which may then be tested in some subsequent study. In response to this comment, we have added a citation to this study design. Hopefully, now you will find it correct.  

Reviewer 3. / Comment 13.

Repetition of previously described things commonly occur in the text. This should be avoided.

RESPONSE: Thank you for this suggestion. We reviewed the abstract and found no unnecessary repetitions. We used academic phrase bank and academic writing style of the University of Manchester. We hope you find this response satisfactory.

Reviewer 3. / Comment 14.

Lines 225-239. Too long description for interaction rate. It does not fit to the methods section of a scientific manuscript. Please write concise and clear sentences to present measures of the current study.

RESPONSE: Thank you for your observation. In response to this comment, we have condensed this paragraph. Hopefully, you will find this response acceptable.

Reviewer 3. / Comment 15.

Lines 242- 270. Bullet points in this part are unnecessary repetitions of previously described measures. There is not any citation for applying these measures.

RESPONSE: Thank you so much for your comment. These are not repeats. In contrast to the previous paragraphs, we introduce new concepts here (interaction rates), which makes it clear how the number of interactions can be compared with the same reach. Hopefully, you will find it satisfactory.

Reviewer 3. / Comment 16.

Procedure of Methods section usually discusses the way of data collection. Statistical analysis/methods is a separate subsection of Methods.

RESPONSE: Thank you for your suggestion. We follow the APA style: “Participants”, “Materials” (“Facebook posts”), “Design” and “Procedure” in the “Method” section. "Procedure" means all methods for study administration, data processing and data analysis. Hopefully, now you will find it correct.

Reviewer 3. / Comment 17.

It is unclear what the Authors regarded as ’control group’.

RESPONSE: Thank you for pointing these out. Smoking cessation support contents were classified into engagement bait, alternative techniques, and control group categories. The control group did not use engagement bait or alternatives techniques. Hopefully, you will find this response acceptable.

Reviewer 3. / Comment 18.

It is unclear how post classification was conducted. Do you mean classifying posts into engagement bait vs alternative techniqus vs control? If yes, why not provide a clear description for classification and allocation into study groups?

RESPONSE: Thank you so much for your comment. We mentioned in the "Design" section that the research is retrospective. The clear description for classification is in the "Design" section (definitions of the categories) and "Procedure" section (the method of the classification and inter-rater reliability). Hopefully, you will find it satisfactory.

Reviewer 3. / Comment 19.

Line 298. ’…the p valuse of less than 0.001 was taken to indicate a highly significant effect.’ Why not calculated effect sizes? It is an incorrect consideration of significance.

RESPONSE: This comment may be a misunderstanding. Two sentences ago it reads that "The effect size was measured by eta squared." Eta squared is a measure of effect size that is commonly used. The value for eta squared was indicated for each result (the last value inside the parentheses): e.g., (c2(2)=6.930, P=.031, h2=0.006); (c2(2)=15.818, P<.001, h2=0.018) or (c2(2)=15.818, P<.001, h2=0.018). Hopefully, you will find this response acceptable.

Reviewer 3. / Comment 20.

The interpretation of the Hungarian language by Facebook was not mentioned in Methods.

RESPONSE: This comment may be a misunderstanding. In the "Participants" ("Methods") section it reads that "We published Hungarian language contents..." Hopefully, now you will find it correct.

Reviewer 3. / Comment 21.

It seems that this Facebook page has a small reach of users. Moreover, ignoring several other factors connected to the operation of the algorithm, it is impossible to draw reliable conclusions in this study.

RESPONSE: Thank you for your suggestion. In our opinion, the following sentence is subjective: "This Facebook page has a small reach of users." The investigated Facebook page had more than 10.000 followers in the study period. A previous cross-sectional study found that two thirds of the followers are smokers. So, it can reach at least 6.000 smokers in the study period. In our opinion, this sample is not small. Other strength is the three-year study period, which allowed us to select 791 Facebook posts out of 1,026 social media contents. As we have published a content almost daily since the "engagement bait" regulation in 2018, this database size seems to be notable. Lastly, the factors of the Facebook Algorithm are shown in the "Introduction" and "Discussion". These factors were constant during the study period. That is why we can analyze these associations in the Facebook Algorithm. Analyzing the Facebook Algorithm is not the first method that we have used. The same method was used in the previously mentioned self-citation (Ref. No. 15.; Facebook Users' Interactions, Organic Reach, and Engagement in a Smoking Cessation Intervention: Content Analysis.). This manuscript and its method have been accepted by the Journal of Medical Internet Research (a top tier journal in the field of health informatics). Of course, if the IJERPH has reservations about the method, we understand that. Hopefully, you will find it satisfactory.

Reviewer 3. / Comment 22.

In Results section, some of the research questions did not meet with the applied statistical methods. E.g., in lines 306-314, a more specific research question would be needed. Here, the Authors explored possible differences between fan group reach vs non-fan group reach of posts categorized in the engagement bait and control groups. Furthermore, presented statistical tests do not support this statement. Without presenting clearly your findings e.g., in a table, these results are questionable.

RESPONSE: Thank you so much for your comment. In response to this comment, this research question was changed: “How do engagement bait sanctions work for smoking cessation support contents on Facebook?” Hopefully, you will find this response acceptable.

Reviewer 3. / Comment 23.

In line 337, the Authors mentioned ’1,000 Facebook users. The exact sample size is unclear. Generally, presenting clearly descriptive characteristics of this study is missing. E.g., presenting in a table the n of engagement bait posts, alternative posts, control posts, n of fan vs non-fan groups, etc. in light of some demographic variables.

RESPONSE: Thank you for your observation. In response to this comment, this sentence was modified in the “Method” section: “We used interaction rates in the current research to express the frequency of the given interaction at the same reach (per one thousand Facebook users).” Hopefully, you will find it satisfactory.

Reviewer 3. / Comment 24.

Some research questions are repeated several times which indicate inadequate approach of study aims.

RESPONSE: Thank you for pointing these out. We have used the University of Manchester academic style of writing, which favors reiterating the research aims and questions in the "Results" and "Discussion" sections. Hopefully, now you will find it correct.

Reviewer 3. / Comment 25.

Discussion and especially Conclusions are not scientifically sound.

RESPONSE: Thank you for your suggestion. We reviewed the "Discussion" and "Conclusions" sections and found no unscientific sentences. If you write a specific instruction, we will correct it immediately. Hopefully, you will find this response acceptable.

We thank you for the time you put in reviewing our paper and look forward to meeting your expectations. We hope that this new version is satisfactory.

Round 2

Reviewer 3 Report

This manuscript has not improved much. More than half of my comments were not adequately responded and I cannot accept superficial responses as well as ignoring corrections in the manuscript.

Methods section is still confusing. Some parts mention an intervention study, while other parts a retrospective study. Please present the methodology of this study clearly. The Readers should understand the main methodology from the manuscript itself without searching the methodology from other articles of the same Authors. Or provide correct citations for article(s) presenting the same study design.

Research questions are still general, that is, does not reflect to this specific Hungarian Facebook page.

I agree with the Authors that this manuscript emphasizes smoking cessation-related posts. However, please consider being prudent when criticizing terminologies. On the one hand, prevention has three levels: primary, secondary, and tertiary. Considering the relatively high adult smoking prevalence rate in Hungary as well as the enormous burden of disease due to smoking in Hungary, I do not think so that we should regard smoking cessation as a primary prevention method in this country. Although the levels of prevention is out of the scope of this manuscript, please consider the meaning of the three levels of prevention. E.g., 'tertiary prevention: stopping progress of an already occurring disease, and preventing complications.' (doi: 10.1016/B978-0-12-415766-8.00002-1) Moreover, all major cessation guidelines consider nicotine and tobacco use as a disease with ICD codes. On the other hand, ‘harm reduction for smokers’ and ‘development of social support skills for non-smokers’ were also mentioned as additional goals of the ‘Cigarette Break’ Facebook page. These belong to different levels of the preventin of smoking.

Lines 106-110: These are the results of another study which does not fit to the structure of Methods section, more closely, to Participants subsection.

If you would present an intervention study, please use the CONSORT statements. However, I think that this study did not aimed to present the results of an intervention. Therefore the content of Participants subsection is questionable.

The citation is still incorrect for the study design (‘hypothesis-generating, retrospective, quantitative content analysis’)

Please reconsider your responses for my previous (Round 1) comments: Reviewer 3. / Comment 2, 6-9, 11-13, 15, 17-18, 20, 23-24. Please provide thorough corrections according to my comments also in the manuscript.

Author Response

Response to Reviewer 3 Comments

We wish to thank you all for your constructive comments. Your comments provided valuable insights to refine its contents. We try to address the issues raised as best as possible.

Reviewer 3. / Comment 1.

This manuscript has not improved much. More than half of my comments were not adequately responded and I cannot accept superficial responses as well as ignoring corrections in the manuscript.

RESPONSE: Thank you to the reviewer for pointing this out. This could be a misunderstanding. We do not want to ignore the corrections in the manuscript! However, we need specific instructions for repairs. Based on general comments such as "The abstract does not reflect clearly the content of the main text." and “Discussion and especially Conclusions are not scientifically sound.”, it is difficult to improve the manuscript. In the first round, we tried to conscientiously respond to specific comments of all the reviewers. We hope you will find this response correct.

Reviewer 3. / Comment 2.

Methods section is still confusing. Some parts mention an intervention study, while other parts a retrospective study. Please present the methodology of this study clearly. The Readers should understand the main methodology from the manuscript itself without searching the methodology from other articles of the same Authors. Or provide correct citations for article(s) presenting the same study design.

RESPONSE: Thank you for this suggestion. Firstly, "intervention study" or "interventional study" expressions are not mentioned in the manuscript. However, the word "retrospective" is mentioned six times in the manuscript. Secondly, an interventional study can be retrospective, these are different approaches and not a contradiction. Interventional studies are those where the researcher intercedes as part of the study design. Additionally, study designs may be classified by the role that time plays in the data collection, either retrospective or prospective. In retrospective studies, the outcome of interest has already occurred, and the data are collected either from records. In response to this comment, the following sentence has been added to the “Method” section: “The current study is retrospective database research where all the events of interest have already happened.” That is why, we used the word "retrospective". Hopefully, you will find this response acceptable.

Reviewer 3. / Comment 3.

Research questions are still general, that is, does not reflect to this specific Hungarian Facebook page.

RESPONSE: Thank you for your valuable observation. We mentioned in the "Limitations and strengths" section that "Content ranking is independent of language barriers." and "Although the contents were in Hungarian, Facebook used the same algorithm it applied for posts in English or any other languages." Because of the general Facebook Algorithm, there is no need for intervention-specific research questions. However, in response to this comment, we modified the research questions again: “How do engagement bait sanctions work for promoting smoking cessation on Facebook?” “What is the relationship between engagement bait techniques used in Facebook-based smoking cessation intervention and users' interactions?” “Are there any differences in Facebook sanctions or users' interactions, if alternative techniques are used instead of engagement bait during a smoking cessation intervention?”. Hopefully, you will find it satisfactory.

Reviewer 3. / Comment 4.

I agree with the Authors that this manuscript emphasizes smoking cessation-related posts. However, please consider being prudent when criticizing terminologies. On the one hand, prevention has three levels: primary, secondary, and tertiary. Considering the relatively high adult smoking prevalence rate in Hungary as well as the enormous burden of disease due to smoking in Hungary, I do not think so that we should regard smoking cessation as a primary prevention method in this country. Although the levels of prevention is out of the scope of this manuscript, please consider the meaning of the three levels of prevention. E.g., 'tertiary prevention: stopping progress of an already occurring disease, and preventing complications.' (doi: 10.1016/B978-0-12-415766-8.00002-1) Moreover, all major cessation guidelines consider nicotine and tobacco use as a disease with ICD codes. On the other hand, ‘harm reduction for smokers’ and ‘development of social support skills for non-smokers’ were also mentioned as additional goals of the ‘Cigarette Break’ Facebook page. These belong to different levels of the prevention of smoking.

RESPONSE: We thank the reviewer for this observation. The term "smoking prevention" is not included in the manuscript. We agree with the reviewer that the levels of smoking prevention are out of the scope of this manuscript. We noticed that the secondary goals of the program (harm reduction, relapse prevention) may be complicated. In response to this comment, the secondary goals of the program have been removed from the manuscript. Hopefully, you will find this response acceptable.

Reviewer 3. / Comment 5.

I Lines 106-110: These are the results of another study which does not fit to the structure of Methods section, more closely, to Participants subsection.

RESPONSE: Thank you so much for your comment. We agree with the reviewer. In response to this comment, these sentences have been removed from the manuscript. Hopefully, you will find this response satisfactory.

Reviewer 3. / Comment 6.

If you would present an intervention study, please use the CONSORT statements. However, I think that this study did not aimed to present the results of an intervention. Therefore the content of Participants subsection is questionable.

RESPONSE: We thank to the reviewer for pointing this out. The current research is not a RCT where the use of CONSORT statements would be expected. The current research is not a traditional (prospective) interventional study. That is why, "intervention study" or "interventional study" expressions are not mentioned in the manuscript. In response to this comment, the following sentences have been added to the “Method” section: "We analyzed a Facebook-based smoking cessation intervention using case-control study method, retrospectively. The contents were also classified retrospectively, not prospectively." Hopefully, you will find it correct.

Reviewer 3. / Comment 7.

The citation is still incorrect for the study design (‘hypothesis-generating, retrospective, quantitative content analysis’).

RESPONSE: Thank you for your comment. In response to this comment, this citation has been removed from the manuscript. Hopefully, you will find it satisfactory.  

Reviewer 3. / Comment 8.

Please reconsider your responses for my previous (Round 1) comments: Reviewer 3. / Comment 2, 6-9, 11-13, 15, 17-18, 20, 23-24. Please provide thorough corrections according to my comments also in the manuscript.

RESPONSE: Thank you for your suggestion. We have considered the responses to Round 1 comments:

  • Comment 2. The abstract does not reflect clearly the content of the main text.
    • Without specific instructions, it is impossible to respond to this comment.
  • Comment 6. Research aims do not consider smoking cessation support on Facebook. Aims are just generally related to Facebook engagement bait. It would be essential to specify study aims and to provide clear and relevant aims.
    • In response to this comment, we modified the research questions again: “How do engagement bait sanctions work for promoting smoking cessation on Facebook?” “What is the relationship between engagement bait techniques used in Facebook-based smoking cessation intervention and users' interactions?” “Are there any differences in Facebook sanctions or users' interactions, if alternative techniques are used instead of engagement bait during a smoking cessation intervention?”.
  • Comment 7. Methods section is the weakest point of this manuscript. In subsection 2.1., the Authors mention an ’intervention’, however, it is unclear why an intervention it is. Moreover, what type of intervention and what is the structure of the intervention?
    • In response to this comment, the following sentences have been added to the “Method” section: "Based on the delivery of health information, this Facebook page is a web-based smoking cessation intervention. Based on the therapeutic method, this intervention is behavioral counseling for smoking cessation with a motivational interviewing approach."
  • Comment 8. In lines 104-107, various smoking prevention approaches were presented. However, in the manuscript, the evidence behind applying these approaches is unclear. Besides, citations seem to be irrelevant self-citations. Please provide correct citations supporting the applied methods.
    • In response to this comment, the following reference has been added to the “Method” section: Lindson N, Thompson TP, Ferrey A, Lambert JD, Aveyard P. Motivational interviewing for smoking cessation. Cochrane Database Syst Rev. 2019;7(7):CD006936. https://doi.org/10.1002/14651858.CD006936.pub4
  • Comment 9. Furthermore, consider that this manuscript targets smoking cessation instead of non-smokers and smokers unwilling to quit completely.
    • In response to this comment, the secondary goals of the program have been removed from the manuscript.
  • Comment 11. Lines 167-196 are unnecessary paragraphs. These could be easily included in the text. Alternatively, the Authors should prepare a table combining and presenting engagement bait and alternative techniques whit examples.
    • In Round 1, we wrote that why this paragraph is necessary. Without new specific instructions, we do not know that what the Reviewer's concerns are about this response.
  • Comment 12. Lines 202-203. Please clarify the study design. Please revise the study design whether it is really a hypothesis-generating study or not. Please provide citation for this study design.
    • In response to this comment, the following sentence has been added to the “Method” section: “This is a hypothesis-generating study, because it explores a set of data searching for relationships and patterns, and then proposes hypotheses which may then be tested in some subsequent study.”
  • Comment 13. Repetition of previously described things commonly occur in the text. This should be avoided.
    • Without specific instructions, it is impossible to respond to this comment.
  • Comment 15. Lines 242- 270. Bullet points in this part are unnecessary repetitions of previously described measures. There is not any citation for applying these measures.
    • In Round 1, we wrote that why this paragraph is necessary. Without new specific instructions, we do not know that what the Reviewer's concerns are about this response.
  • Comment 17. It is unclear what the Authors regarded as ’control group’.
    • The Reviewer explains that something is missing from the manuscript. However, this was mentioned in the original submission: "The control group consisted of 375 Facebook posts, which did not use engagement bait or alternatives techniques." Without specific instructions, we do not know that what the Reviewer's concerns are about this.
  • Comment 18. It is unclear how post classification was conducted. Do you mean classifying posts into engagement bait vs alternative techniques vs control? If yes, why not provide a clear description for classification and allocation into study groups?
    • In Round 1, we wrote that how post classification was conducted. Without new specific instructions, we do not know that what the Reviewer's concerns are about this response. Earlier in "Comment 11", the Reviewer wrote that a description of the classification was unnecessary. However, "Comment 18" confirms to us that descriptions illustrating the classification should be left in the manuscript.
  • Comment 20. The interpretation of the Hungarian language by Facebook was not mentioned in Methods.
    • This is not true. The Reviewer explains that something is missing from the manuscript. However, this was mentioned in the original submission ("Methods" section): "We published Hungarian language contents…" In Round 1, we mentioned this misunderstanding (similarly to "effect size"). It is incomprehensible what the Reviewer wants with this in the Round 2.
  • Comment 23. In line 337, the Authors mentioned ’1,000 Facebook users. The exact sample size is unclear. Generally, presenting clearly descriptive characteristics of this study is missing. E.g., presenting in a table the n of engagement bait posts, alternative posts, control posts, n of fan vs non-fan groups, etc. in light of some demographic variables.
    • In Round 1, we modified the text based on this comment. Without new specific instructions, we do not know that what the Reviewer's concerns are about this response. Earlier in "Comment 15", the Reviewer wrote that a description of interaction rates was unnecessary. However, "Comment 23" confirms to us that descriptions illustrating the interaction rates should be left in the manuscript.
  • Comment 24. Some research questions are repeated several times which indicate inadequate approach of study aims.
    • In Round 1, we mentioned which academic style was used. Without new specific instructions, we do not know that what the Reviewer's concerns are about this response.

Hopefully, now you will find it acceptable.  

We thank you for the time you put in reviewing our paper and look forward to meeting your expectations. We hope that this new version is satisfactory.